# How to account for the uncertainty from standard toxicity tests in species sensitivity distributions: An example in non-target plants

**Sandrine Charles**[1]*, **Dan Wu**[1], **Virginie Ducrot**[2]

**1** Univ Lyon, Université Lyon 1, UMR CNRS 5558, Villeurbanne, France, **2** Bayer AG, Crop Science, Monheim, Germany

* sandrine.charles@univ-lyon1.fr

**Data Availability Statement:** All data files are available from the ZENODO database (DOI: 10. 5281/zenodo.3906705).

## Abstract

This research proposes new perspectives accounting for the uncertainty on 50% effective rates ($ER_{50}$) as interval input for species sensitivity distribution (SSD) analyses and evaluating how to include this uncertainty may influence the 5% Hazard Rate ($HR_5$) estimation. We explored various endpoints (survival, emergence, shoot-dry-weight) for non-target plants from seven standard greenhouse studies that used different experimental approaches (vegetative vigour vs. seedling emergence) and applied seven herbicides at different growth stages. Firstly, for each endpoint of each study, a three-parameter log-logistic model was fitted to experimental toxicity test data for each species under a Bayesian framework to get a posterior probability distribution for $ER_{50}$. Then, in order to account for the uncertainty on the $ER_{50}$, we explored two censoring criteria to automatically censor $ER_{50}$ taking the $ER_{50}$ probability distribution and the range of tested rates into account. Secondly, based on dose-response fitting results and censoring criteria, we considered input $ER_{50}$ values for SSD analyses in three ways (only point estimates chosen as $ER_{50}$ medians, interval-censored $ER_{50}$ based on their 95% credible interval and censored $ER_{50}$ according to one of the two criteria), by fitting a log-normal distribution under a frequentist framework to get the three corresponding $HR_5$ estimates. We observed that SSD fitted reasonably well when there were at least six distinct intervals for the $ER_{50}$ values. By comparing the three SSD curves and the three $HR_5$ estimates, we shed new light on the fact that both propagating the uncertainty from the $ER_{50}$ estimates and including censored data into SSD analyses often leads to smaller point estimates of $HR_5$, which is more conservative in a risk assessment context. In addition, we recommend not to focus solely on the point estimate of the $HR_5$, but also to look at the precision of this estimate as depicted by its 95% confidence interval.

## Introduction

Today, Species Sensitivity Distributions (SSD) are established as a key tool for the environmental risk assessment (ERA) of chemicals [1, 2]. They provide a reliable assessment of the range of sensitivities within a plant or animal community of interest [3] and thereby allow to

**Funding:** The European Crop Protection Association (ECPA, https://www.ecpa.eu) financially supported this work. The funder provided support in the form of salaries for authors [DW], but did not have any additional role in the study design, data collection and analysis, decision to publish, or preparation of the manuscript. The specific roles of these authors are articulated in the 'author contributions' section. One of the author [VD] is employed by a commercial company (Bayer AG, Crop Science). This author collected and made available the raw data used for the modelling work and participated to the preparation of the manuscript but had no role in the study design, data analysis and data interpretation.

**Competing interests:** One of the author [VD] is employed by a commercial company (Bayer AG, Crop Science). This author collected and made available the raw data used for the modelling work and participated to the preparation of the manuscript but had no role in the study design, data analysis and data interpretation. This does not alter our adherence to PLOS ONE policies on sharing data and materials.

estimate indicators such as the 5% hazard concentration or rate ($HC_5$ or $HR_5$) that is the hazardous concentration or rate prone to affect 5% of the species within the community. An estimation of the $HC_5$ or $HR_5$ can be obtained from the fit of a probability distribution on a collection of toxicity values, such as 50% effective concentrations or rates ($ER_{50}$ or $ER_{50}$), thus requiring performing a statistically robust analysis. Toxicity values are usually derived from a regression model fitted on toxicity test data observed at several treatment levels at a target time point. This fit provides toxicity values as point estimates, but an uncertainty can also be associated to them, either through a confidence interval (under a frequentist framework) or a credible interval (under a Bayesian framework). Nevertheless, this uncertainty, as well as other sources of uncertainty [4], is rarely accounted for in $HC_5$ or $HR_5$ estimates afterwards. This motivated our study, supported by recent works that also recognise the usefulness of considering interval ecotoxicological data [5].

The SSD method is largely used in the field of non-target terrestrial plant (NTTP) studies for the purpose of assessing the risk of plant protection products [6]. NTTP are defined as non-crop plants located outside the treatment area according to the Guidance Document (GD) on Terrestrial Ecotoxicology [6]. In the case of NTTP studies, treatment levels or exposure concentrations are rather called `tested rates`, corresponding to application rates in field. Subsequently, we will use the notations $ER_{50}$ and $HR_5$ hereafter.

For the SSD analyses, up to ten NTTP species are usually studied for the ERA of pesticides. The used tested rates are selected prior to the experiments, sometimes being the same for some of or all the chosen species. The highest tested rate usually corresponds to the highest authorised application rate of the herbicide in the field, which ensures the realism of the ecotoxicological evaluation towards agricultural practices. However, this highest tested rate might be too low to elicit large toxic effects (*i.e.*, close to 100% effect, as it is classically done when setting the highest dose for a dose-response analysis for animals) especially for fungicides and insecticides. This specific point will be discussed in our paper. Consequently, unbounded right-censored $ER_{50}$ values (namely $ER_{50}$ greater than the highest tested rate) can occur when the range of tested rates does not really match the observed sensitivity of a plant species or when this species is not affected at the highest tested rate intended according to good agricultural practices. Note that such unbound $ER_{50}$ values may have been produced using a validated standard experimental procedure, so that there is no reason to question them. Additionally, the GD on Terrestrial Ecotoxicology does not provide any advice on how to deal with unbound $ER_{50}$ values or with the uncertainty associated with the $ER_{50}$ estimates when performing SSD analyses [6]. As a consequence, the common practice is first to ignore the uncertainty by considering point estimates only (usually the mean estimate), and second, either to discard unbound $ER_{50}$ values from the analysis or to substitute them with arbitrary values (*e.g.*, the highest tested rate), even if rarely done in practice. Nevertheless, performing in such a way is a clear loss of valuable information with some drawbacks. Ignoring uncertainty prevents to account for low (*i.e.*, in the lower tail of their probability distribution) or high (*i.e.*, in the upper tail of their probability distribution) $ER_{50}$ values that are statistically probable, thus leading to potentially biased $HR_5$ estimates (*i.e.*, either over- or under-estimated values). Discarding unbound $ER_{50}$ values may derive in (i) a range of remaining $ER_{50}$ values that may not cover the full range of sensitivities as originally displayed in the set of the chosen species (the most sensitive or the most tolerant species may for example be excluded, thus producing biased $HR_5$ estimates, either over- or under-estimated); (ii) unbound $ER_{50}$ values can occur for many species, so that, after discarding them, the small sample size of the input data set might then not be sufficient to allow an SSD analysis to be performed. This latter issue is of great concern for risk assessment, since the SSD analysis is currently the only higher tier option prescribed by the GD on Terrestrial Ecotoxicology and widely accepted by authorities. Thus, being unable to finalise an

SSD analysis may prevent refining the risk assessment of some chemical substances. The GD on Terrestrial Ecotoxicology indeed considers that the SSD analysis is more suitable than a tier-1 approach (based on a single endpoint and a single species) to achieve the environmental protection goal because it takes into account the available data on the sensitivity of several species [6]. Moreover, substituting unbounded $ER_{50}$ with arbitrary values would be a fairly arbitrary way that does not make much sense and leads to the possibility to produce biased $HR_5$ estimates subjectively.

Within this context, based on seven NTTP case studies, each including several data sets, we aimed to revisit SSD analyses by accounting for both the uncertainty on $ER_{50}$ values (referred to as interval-censored values hereafter) together with the inclusion of censored values, in particular right-censored values (corresponding to unbounded $ER_{50}$ values) what commonly happens with toxicity tests in practice for NTTP. Indeed, left-censored values are rare because the tested rate range as imposed by the standard protocols is better adapted to assess effects for the more sensitive species. We also tried to quantify how both types of censored values may influence the final estimate of the $HR_5$.

## Materials and methods

### Materials

Laboratory experiment data sets on NTTP were available for seven case studies on products with various herbicidal mode of action (Table 1). Each study provides data from two toxicity tests: seedling emergence (SE) according to OECD guideline 208 [7] and vegetative vigour (VV) according to OECD guideline 227 [8]. For each study, 10 species (thereafter named using their EPPO code [9]; see S1 Table for common names of species) were exposed to a range of five tested rates of a product plus a control (*i.e.*, absence of product), which were applied either to the soil surface (SE tests) or directly to the plants (VV tests). Besides, in study 4, extra experiments at lower tested rates were conducted for two of the species (CUMSA and LYPES) in the VV test, and for study 7, extra experiments at lower tested rates for two of the species (ALLCE and BEAVA) were carried out in the SE test.

**Table 1. Brief description of the seven studies.**

| Study | Product[1] | Tested species (EPPO code)[2] | Tested rate unit |
|---|---|---|---|
| study 1 | product 1 | ALLCE AVESA BEAVA BRSNW CUMSA GLXMA HELAN LYPES TRZAW ZEAMA | ml product/ha |
| study 2 | product 2 | ALLCE AVESA BEAVA BRSNW CUMSA GLXMA HELAN LOLPE LYPES ZEAMA | g a.s./ha |
| study 3 | product 3 | ALLCE BEAVA BRSNW CUMSA FAGES GLXMA LOLPE LYPES TRZAW ZEAMA | ml product/ha |
| study 4 | product 4 | ALLCE AVESA BEAVA BRSNW CUMSA GLXMA HELAN LYPES TRZAW ZEAMA | ml product/ha |
| study 5 | product 5 | ALLCE AVESA BEAVA BRSNW CUMSA GLXMA HELAN LOLPE LYPES ZEAMA | g a.s./ha |
| study 6 | product 6 | ALLCE AVESA BEAVA BRSNW CUMSA GLXMA HELAN LYPES TRZAW ZEAMA | g product/ha |
| study 7 | product 7 | ALLCE AVESA BEAVA BRSNW CUMSA GLXMA HELAN LYPES TRZAW ZEAMA | g product/ha |

[1] See S2 Table for formulations of active substances of the seven products.

[2] EPPO: European and mediterranean Plant Protection Organization; see S1 Table for corresponding species and common names to the EPPO code.

During experiments, plants were observed for 21 days after day 0. Day 0 is defined as the day at which 50% of the control seedlings have emerged for SE tests and as the day of application for VV tests. During the 21-day observation period, seedling emergence, seedling survival and visual injury in each replicate were followed weekly (at days 0, 7, 14 and 21) in SE tests, while plant survival and visual injuries were followed weekly in VV tests, also in each replicate. At the end of the experiments (in both SE and VV tests), shoots were cut-off and dried up, then the shoot dry weight was measured in each replicate. For each study, five quantitative endpoints at day 21 were thus available: emergence, survival and shoot dry weight for SE tests, survival and shoot dry weight for VV tests.

## Methods

To assess the effects of the studied herbicides on NTTP, we first analysed the effects of the seven products on the five endpoints for each of the 10 species (that is a total of $7 \times 5 \times 10$ data sets) by fitting a dose-response model to experimental toxicity test data thus getting $ER_{50}$ estimates for each data set. The modelling process was carried out under a Bayesian framework, which ensures to get a posterior probability distribution for the $ER_{50}$ which can then be used as a basis to quantify the uncertainty on the $ER_{50}$. Then, these $ER_{50}$ values, also considering their uncertainty, were used as inputs for the SSD analyses leading to the $HR_5$ estimates.

**Dose-response model.**   For SE tests, observed data at day 21 for replicate $i$ can be described as $(R_i, N_i^{init}, N_i^{emer}, N_i^{surv}, W_i)$, where $R_i$ is the tested rate, $N_i^{init}$ the number of sown seeds, $N_i^{emer}$ the number of emerged seedlings, $N_i^{surv}$ the number of surviving seedlings and $W_i$ the shoot dry weight of surviving seedlings. For VV tests, observed data at day 21 for replicate $i$ can be described as $(R_i, N_i^{init}, N_i^{surv}, W_i)$, where $R_i$ is the tested rate, $N_i^{init}$ the number of initial plants, $N_i^{surv}$ the number of surviving plants and $W_i$ the shoot dry weight of surviving plants.

The number of emerged seedlings (SE test) and the number of surviving seedlings or plants (SE and VV tests) follow a binomial distribution, with an emergence probability (resp. a survival probability) as a function of the tested rate (see Eqs (1), (2) and (3)):

$$N_i^{emer} \sim \mathcal{B}(N_i^{init}, f(R_i)) \tag{1}$$

$$N_i^{surv} \sim \mathcal{B}(N_i^{emer}, f(R_i)) \tag{2}$$

$$N_i^{surv} \sim \mathcal{B}(N_i^{init}, f(R_i)) \tag{3}$$

Assuming that $W_i$ is normally distributed with mean $\mu_i$ and standard deviation $\sigma$, with $\mu_i$ defined as a function of the tested rate, we get:

$$W_i \sim \mathcal{N}(f(R_i), \sigma^2) \tag{4}$$

In Eqs (1) to (4), $f$ was chosen as three-parameters log-logistic function:

$$f(x) = \frac{d}{1 + \left(\frac{x}{e}\right)^b} \tag{5}$$

Parameters $b$, $d$ and $e$ are positive. Parameter $b$ is a shape parameter translating the intensity of the effect, $d$ corresponds to the endpoint in control data (*i.e.*, in absence of product) and $e$ corresponds to the $ER_{50}$. Within the Bayesian framework, we have to specify a prior distribution for model parameters $b$, $d$, $e$ (and $\sigma$ in case of modelling shoot dry weight data). The prior distributions are given in Table 2.

**Table 2. Specification of prior distributions for model parameters.**

| Emergence or survival | | Shoot dry weight | |
|---|---|---|---|
| Parameter | Prior distribution | Parameter | Prior distribution |
| $log_{10} b$ | $\mathcal{U}(-2, 2)$ | $log_{10} b$ | $\mathcal{U}(-2, 2)$ |
| $d$ | $\mathcal{U}(0, 1)$ | $d$ | $\mathcal{U}(0, d_{max})^1$ |
| $log_{10} e$ | $\mathcal{N}(\mu, \sigma)^2$ | $log_{10} e$ | $\mathcal{N}(\mu, \sigma)^2$ |
| | | $\sigma$ | $\mathcal{U}(0, d_{max}/2)^1$ |

[1] $d_{max}$ equals twice the highest observed shoot dry weight for the species under consideration. The observation with the highest observed shoot dry weight is excluded from the data set before running inference.

[2] $\mu = \frac{log_{10}(maxR) + log_{10}(minR)}{2}$ and $\sigma = \frac{log_{10}(maxR) - log_{10}(minR)}{4}$, where $minR$ and $maxR$ are the lowest and the highest tested rates, respectively.

*Estimation of parameters.* Model computations were performed in R [10] with JAGS using Gibbs sampling via Markov Chain Monte Carlo (MCMC) simulations [11]. The R-package `morse` [12] was used to analyse emergence and survival data. In package `morse`, if no inhibition of plant emergence (or if no survival) is observed in control groups, parameter *d* is automatically set to 1 by default. Hence a two-parameters log-logistic model is fitted to the data. Emergence and survival data can also be practically analysed with the MOSAIC platform [13]. A *modus operandi* is provided in S1 Appendix. A specific R-code based on the R-package `rjags` was built to fit shoot dry weight data. This code is made freely available through an R-shiny web application (https://mosaic.univ-lyon1.fr/growth), for reproducibility of the results for shoot dry weight data.

Three chains were run firstly for an initialisation phase of 3000 iterations and a burn-in phase of 5000 iterations, then Ratery and Lewis's Diagnostic was used to set the necessary thinning and the number of iterations to reach a given level of precision in posterior samples. These posterior samples allow to get a joint posterior probability distribution as well as marginal posterior probability distributions for all model parameters.

**Censoring $ER_{50}$ estimates to account for the uncertainty.** The output of interest from the previous dose-response analyses consists of the posterior probability distribution of the $ER_{50}$ (Fig 1A) allowing to quantify the uncertainty on the $ER_{50}$ estimation summarised as a 95% credible interval (*CI95*), representing the range of values within which the $ER_{50}$ has 95% of chance to be found. Hence, we considered the use of *CI95* of $ER_{50}$ estimates as inputs of SSD analyses, instead of point estimates (median or mean values), as a good way to account for the uncertainty on the $ER_{50}$ estimates into subsequent analyses. But the following questions then arise: should we always consider the bounds of *CI95* as the uncertainty limits of the $ER_{50}$

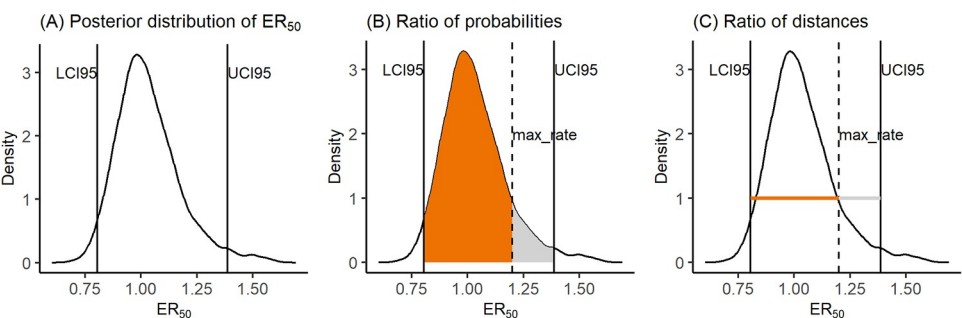

**Fig 1.** Example of posterior probability distribution of $ER_{50}$ (A) and calculation of both censoring criteria (B-C).

and use this interval as an $ER_{50}$ input value for the SSD analysis? Is the $ER_{50}$ estimate always precise enough to be used as it is in the SSD compared to the range of tested rates? How to account for the precision of the $ER_{50}$ estimate regarding the range of tested rates? Is there any situation for which the estimated $ER_{50}$ should be considered as right-censored?

To ensure agronomic realism, the common practice in standard toxicity tests with NTTP is to use the maximal field application rate as the maximum tested rate in the experimental design. However, some species are not affected or only slightly affected at the highest intended application rate: the application rates that would be needed to create high effects in the plants are unknown, thus leading to unbounded values of $ER_{50}$ estimates (greater than the highest tested rate). Such $ER_{50}$ estimates will usually be not precise (having a large $CI95$) which may suggest to rather consider them as a right-censored value. That is why we carefully considered the relevance of the $ER_{50}$ estimates (quantified through their $CI95$) regarding the range of tested rates, in particular the highest tested rate (*max_rate*). In order to decide on the most appropriate mathematical option for automatically right-censoring the $ER_{50}$, we propose two criteria based on overlapping ratios between [$LCI95;max\_rate$] and [$LCI95;UCI95$] intervals, where $LCI95$ and $UCI95$ are the lower and upper bounds of the $CI95$, respectively:

1. A first criterion based on a ratio of probabilities (denoted $C_1$, Eq (6)) defined as the ratio of the probability that the $ER_{50}$ lies within [$LCI95;max\_rate$] over the probability that the $ER_{50}$ lies within [$LCI95;UCI95$]; as illustrated on Fig 1B, criterion $C_1$ is calculated as the ratio of the orange surface divided by the (orange + grey) surface.

$$C_1 = \frac{P(LCI95 \leq ER_{50} \leq max\_rate)}{P(LCI95 \leq ER_{50} \leq UCI95)} \tag{6}$$

2. A second criterion based on a ratio of distances (denoted $C_2$, Eq (7)) defined as the ratio of the distance *max_rate* minus $LCI95$ (if *max_rate* < $LCI95$, then the distance is set to 0) over the extend of the $CI95$; as illustrated on Fig 1C, criterion $C_2$ is calculated as the ratio of the orange segment divided by the (orange + grey) segment.

$$C_2 = \frac{max\_rate - LCI95}{UCI95 - LCI95} \tag{7}$$

*Decision*. Once the criterion is calculated, we need a decision threshold (denoted T) to right-censor or not the $ER_{50}$. If the ratio is greater than T, we keep an interval-censored $ER_{50}$ corresponding to the whole $CI95$; otherwise, we consider a right-censored $ER_{50}$ with a lower bound being the minimum between $LCI95$ and *max_rate* (Eq (8)):

$$\text{censored } ER_{50} = \begin{cases} [LCI95, UCI95] & \text{if } ratio > T \\ [min(LCI95, max\_rate), +\infty) & \text{if } ratio \leq T \end{cases} \tag{8}$$

**SSD analyses.** Our final objective is to explore the influence of considering the uncertainty on $ER_{50}$ in SSD analyses and specifically its impact on $HR_5$ estimates. Given the way we have taken the uncertainty on $ER_{50}$ into account (see above), this means studying how interval- and/or right-censored $ER_{50}$ input values impact the SSD analysis and the $HR_5$ estimation. Thus, SSD analyses were carried out based on $ER_{50}$ values coming from the seven studied firstly analysed with a dose-response model as previously described. For each case study and

each endpoint, based on dose-response fitting results, we considered input $ER_{50}$ values for an SSD analysis in the different following ways:

i.   only point estimates (chosen as the medians of the probability distributions of the $ER_{50}$ estimates);

ii.  interval-censored $ER_{50}$ based on their $CI95$, and we used these intervals as such in a mathematically sound way;

iii. censored $ER_{50}$ according to criterion 1 with a decision threshold T = 0.5 (denoted C1T0.5), and we used these censored $ER_{50}$ as such in a mathematically sound way.

SSD analyses were run by fitting a log-normal probability distribution to $ER_{50}$ input values under a frequentist framework based on the R-package `fitdistrplus` [14]. This R-package allows the user to deal with censored data in a mathematically sound way. An alternative way is to use the web platform MOSAIC and its SSD module https://mosaic.univ-lyon1.fr/ssd [15].

## Results

### Dose-response analyses

All results on dose-response analyses are displayed in files `report_xxx.pdf` in S1 Archive for each case study, each species and each endpoint (five files per case study). Under a Bayesian framework, whatever the data set, the species and the endpoint, we always succeeded in fitting a dose-response curve and getting a posterior probability distribution on the $ER_{50}$. For certain endpoints in certain studies, we got a well-shaped sigmoidal dose-response curve with a median estimate of the $ER_{50}$ within the range of tested rates. Nevertheless, in cases where the herbicides did not elicit a strong effect on the chosen species, we got a flat dose-response curve with a high median estimate of the $ER_{50}$, in particular for the survival endpoint of the VV tests.

Fig 2 illustrates an example of a dose-response curve along with some goodness-of-fit criteria. The data we used for this example is the shoot dry weight of the VV test from case study 1 for species BEAVA. The median fitted dose-response curve in Fig 2A is represented by a solid orange line associated with its $CI95$ as a grey band; it describes the shoot dry weight of the sugar beet as a function of the product tested rate. The goodness-of-fit for the fitted model can be checked using posterior predictive check (PPC) plot illustrated in Fig 2B. The PPC plot shows the observed shoot dry weight values against their corresponding shoot dry weight predictions (black dots), along with their $CI95$ (vertical segments, green if the $CI95$ contains the observed value and red otherwise). The model is considered to be well fitted if around 95% of black dots are within $CI95$. Please note that for the emergence and survival datasets, the previous statement is not necessarily expected because observations are pooled per tested rate. The precision of the model parameter estimates can be visualised in Fig 2C by comparing the posterior distribution (orange surface) to the prior one (grey surface) for each parameter; in Fig 2D, we can check for correlations between parameters. A narrower posterior distribution compared to the prior one for each parameter and low correlations between parameters are expected to ensure the goodness-of-fit of the model; that is the case in this example.

### Censoring on $ER_{50}$ estimates

The censoring decision for an $ER_{50}$ depends on both the criterion ($C_1$ or $C_2$) and the decision threshold T. To study the influence of the criterion and the decision threshold on censoring decisions, we tried seven T values: 0, 0.2, 0.4, 0.5, 0.6, 0.8, 1.0, with each criterion. Fig 3 provides an example of censored $ER_{50}$ values obtained according to both criteria and the seven

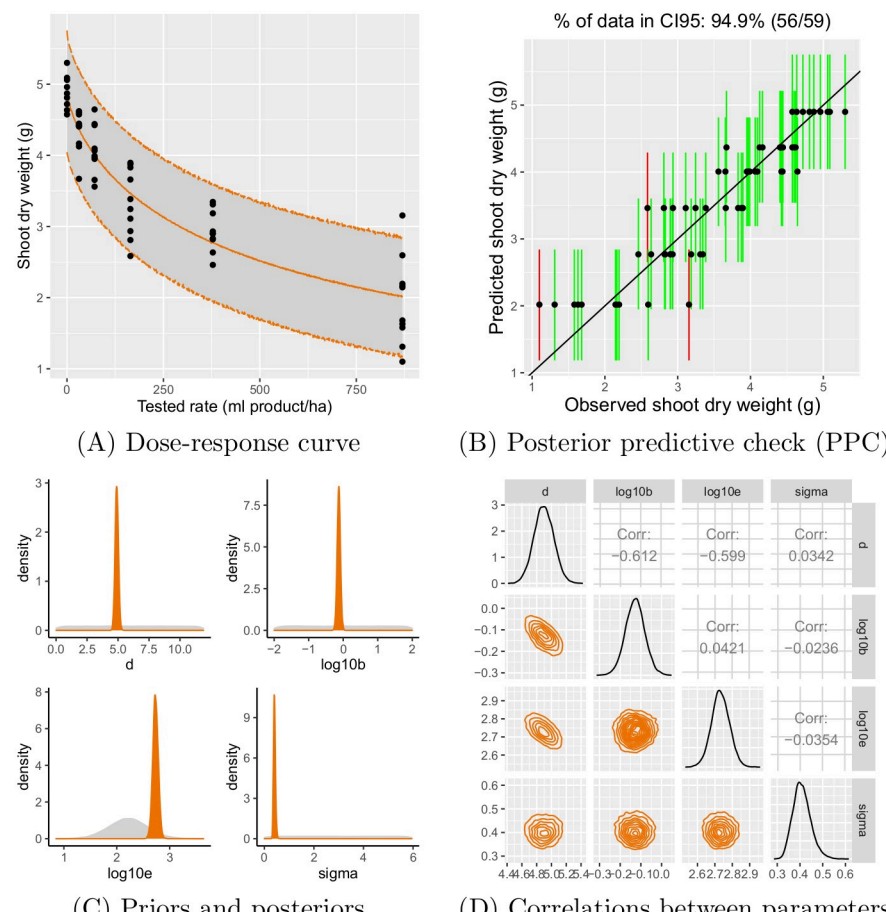

(A) Dose-response curve

(B) Posterior predictive check (PPC)

(C) Priors and posteriors

(D) Correlations between parameters

**Fig 2.** Example of a dose-response curve (A), posterior predictive check (B), prior and posterior distributions of parameters (C) and correlations between parameters (D).

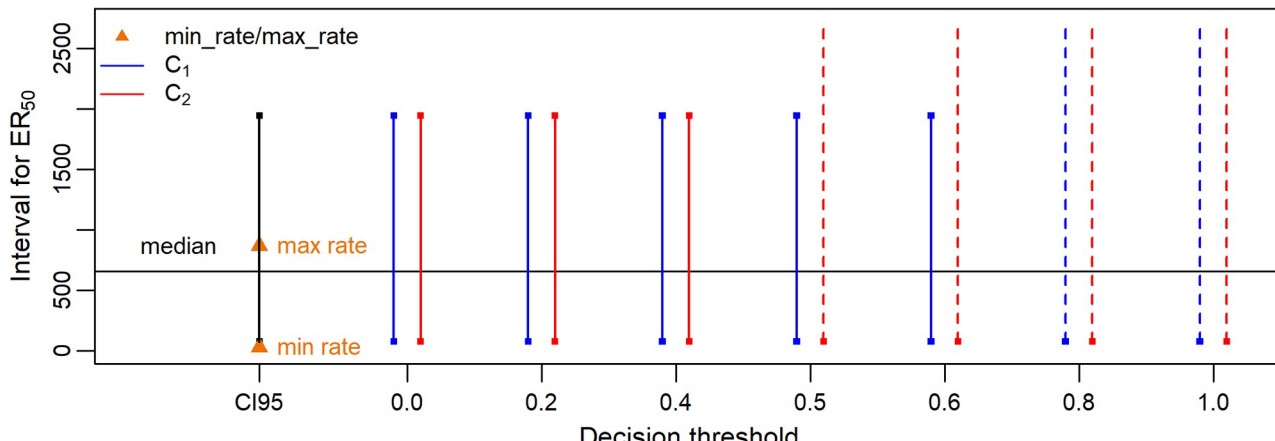

**Fig 3. Censored $ER_{50}$ according to both criteria and the seven decision thresholds for the shoot dry weight endpoint of the VV test from case study 1—species ALLCE.** The two orange triangles stand for the lowest and the highest tested rates. Vertical segments of three different colours (black, blue, red) represent the $CI95$ of $ER_{50}$, the censored $ER_{50}$ according to $C_1$ and the censored $ER_{50}$ according to $C_2$, respectively; solid vertical segments are for bounded intervals while dotted vertical segments stand for right-unbounded intervals; the black horizontal line represents the median of the $ER_{50}$ estimate.

decision thresholds. The data we used for this example is the shoot dry weight of the VV test from case study 1 for species ALLCE.

In this example, the $ER_{50}$ is either interval-censored or right-censored depending on the criterion and the T value. Moreover, censored $ER_{50}$ values vary slightly according to the criterion and the decision threshold. Most of the time, criteria $C_1$ and $C_2$ lead to the same censoring decision for our seven case studies, thus criterion $C_1$ (based on the whole probability distribution of the $ER_{50}$) was finally preferred. See files ER50_censoring.pdf in S1 Archive (seven files in total) for results on other species and other endpoints. Regarding the decision threshold T, in the following cases, the seven decision thresholds led to the same censoring decision:

- case 1: when the $CI95$ of the $ER_{50}$ is utterly within the range of tested rates, an $ER_{50}$ interval-censored by its $CI95$ is always obtained;

- case 2: when the $CI95$ of the $ER_{50}$ is utterly out of the range of tested rates, a right-censored $ER_{50}$ [$max\_rate$, $+ \infty$] is always obtained.

Consequently, the decision threshold influences the censoring decision when there is an overlap between the $CI95$ of the $ER_{50}$ and the range of tested rates. In this case, the higher T is, the more often we will decide to right-censor the $ER_{50}$. Hence, in certain cases, a too high T value may generate a lot of right-censored $ER_{50}$ values and lead to consider some $ER_{50}$ estimates right-censored while we would have rather preferred to use their $CI95$ to quantify their uncertainty. On the other hand, a too low T value may almost always lead to decide to use an interval-censored $ER_{50}$ with its $CI95$, even in cases where most of the possible values for the $ER_{50}$ estimate within the support of its posterior probability distribution are greater than the highest tested rate; so, in such a case, we would have rather considered to right-censor it regarding the range of the tested rates. Therefore, we have a preference for T = 0.5, as a neutral value. Hence, for the subsequent SSD analyses, we considered only censored $ER_{50}$ values according to $C_1$ and T = 0.5 (C1T0.5).

## SSD and $HR_5$

Three ways of handling $ER_{50}$ values in SSD analyses were studied and compared for the seven case studies. For each case study, all results on SSD and $HR_5$ are displayed in files SSD_ana-lyses.pdf in S1 Archive (seven files in total). In total, we did SSD analyses on 105 data sets (7 studies × 5 endpoints × 3 types of $ER_{50}$). We had almost no convergence problem for parameter estimation, except for the survival and shoot dry weight endpoints of the VV test for case studies 2 and 5. In case study 2, for the survival endpoint, fitting a log-normal distribution to the data set with censored $ER_{50}$ values according to C1T0.5 failed because the 10 censored $ER_{50}$ values were in fact equal to the same interval [$max\_rate$, $+ \infty$]. In other cases, convergence failed because all censored $ER_{50}$ values were too close from each other.

Fig 4 illustrates an example of three SSD analyses based only on medians of $ER_{50}$ (A), $ER_{50}$ interval-censored by their $CI95$ (B) and $ER_{50}$ censored according to C1T0.5 (C). The obtained $HR_5$ estimates are denoted by $HR_{5,1}$, $HR_{5,2}$ and $HR_{5,3}$, respectively. The data for this example is the shoot dry weight of the VV test from case study 1. In this example, the three SSD curves fitted well to $ER_{50}$ values. The SSD curve in Fig 4C has a larger 95% confidence interval ($CoI95$) than the ones in Fig 4A and 4B. Estimated $HR_{5,3}$ was smaller than $HR_{5,1}$ and $HR_{5,2}$ with a larger $CoI95$.

**Summary from the seven case studies.** Results on $HR_5$ based on the three ways of handling $ER_{50}$ values for the seven data sets are given in Table 3 and corresponding SSD curves are given in files SSD_analyses.pdf in S1 Archive.

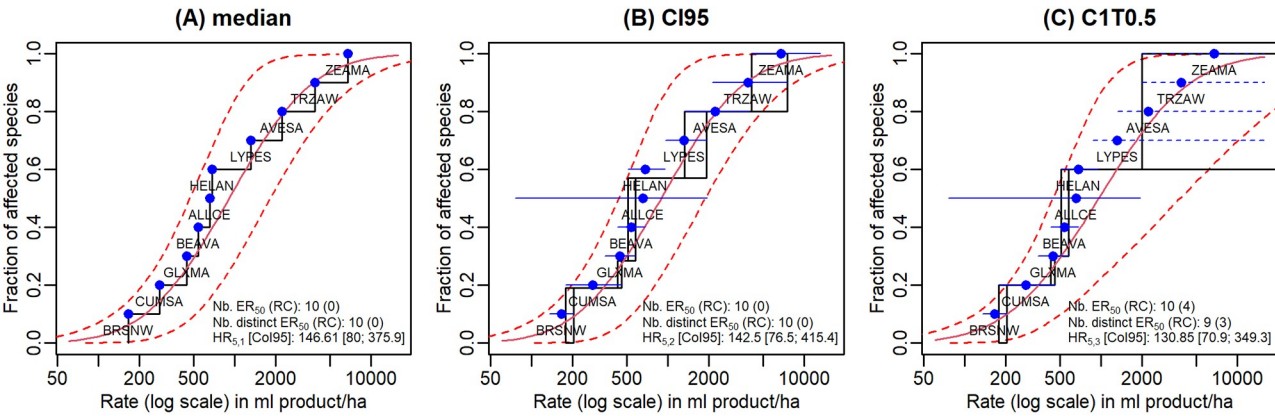

**Fig 4. Example of three SSD analyses based respectively on $ER_{50}$ medians, $CI95$ of $ER_{50}$ and censored $ER_{50}$ according to C1T0.5 for the shoot dry weight endpoint of the VV test from case study 1.** Blue points represent fractions of affected species (EPPO code) ordered by the median of their $ER_{50}$. Solid blue horizontal segments stand for interval-censored $ER_{50}$ by their $CI95$ and dotted ones stand for right-censored $ER_{50}$. Solid red curves represent fitted SSD curves and dotted ones represent 95% confidence interval ($CoI95$) of the fitted SSD curves. Meaning of legends: Nb. $ER_{50}$ (RC) stands for the number of $ER_{50}$ (the number of right-censored $ER_{50}$); Nb. distinct $ER_{50}$ (RC) stands for the number of distinct $ER_{50}$ (the number of distinct right-censored $ER_{50}$); $HR_5[CoI95]$ stands for the estimated $HR_5$ with its $CoI95$.

According to SSD curves, SSD were poorly fitted when there were less than six distinct intervals for the $ER_{50}$ values among the 10 input intervals, most of which being right-censored (*e.g.*, the emergence endpoint for case studies 1, 2, 3, 6 and 7; the survival endpoint of the SE test for case studies 2 and 3; the survival endpoint of the VV test for case studies 1, 3 and 5; the shoot dry weight endpoint of the VV test for case study 5). In such cases, we often found that $HR_5$ estimates were imprecise with a large $CoI95$. The $HR_{5,3}$ estimates taking the right-censoring into account were often greater than the $HR_{5,1}$ and $HR_{5,2}$ estimates based on only medians of $ER_{50}$ or based on interval-censored $ER_{50}$ by their $CI95$. Please note that we performed these SSD analyses anyway to obtain $HR_5$ estimates and to illustrate that $HR_5$ values obtained for data sets where less than six species show clear adverse effects are not precise. Therefore, it would make no sense to fit an SSD in such cases for routine analyses.

According to SSD curves, SSD fitted reasonably well when there were at least six distinct intervals for $ER_{50}$ values as inputs, only some of which being right-censored (*e.g.*, the emergence endpoint for case study 5; the survival endpoint of the SE test for case studies 1, 4, 5, 6 and 7; the shoot dry weight endpoint of the SE test for the seven case studies; the survival endpoint of the VV test for case studies 4, 6 and 7; the shoot dry weight endpoint of the VV test for case studies 1, 3, 4, 6 and 7). We almost always observed that the $HR_{5,3}$ were lower than the $HR_{5,1}$ and $HR_{5,2}$, except for the survival endpoint of the SE test for case study 4, where we had a $HR_{5,3}$ twice greater than the $HR_{5,1}$ and $HR_{5,2}$; for the three endpoints of the SE test for case study 5, $HR_{5,1}$, $HR_{5,2}$ and $HR_{5,3}$ were very close and for the shoot dry weight endpoint of the VV test for case study 7, the $HR_{5,3}$ was a little greater than the $HR_{5,1}$ but less than $HR_{5,2}$.

Concerning the influence of including right-censored data or not on the $HR_5$ estimate, based on SSD curves for our seven case studies, we found that for cases with an $HR_{5,3}$ lower than the $HR_{5,1}$ and $HR_{5,2}$ estimates, the right-censored $ER_{50}$ values were rather obtained for less sensitive species (*i.e.*, species having higher median $ER_{50}$ values). This result was found frequently among the seven case studies and most of the time in case studies for which there were few right-censored $ER_{50}$ values for measured endpoints; this reflects that such a situation will occur in most of the cases encountered when analysing SSD data for NTTP. This comes from the fact that the tested rate range (as imposed by the standard protocols) is better adapted to assess effects on the more sensitive species. On the other hand, we found that for certain cases

**Table 3. Comparison of results on $HR_5$ based on different handling criteria for input $ER_{50}$ values.**

| Study | Endpoint | Median | | CI95 | | C1T0.5 | | |
|---|---|---|---|---|---|---|---|---|
| | | Nb.(RC)[1] | $HR_{5,1}$ CoI95][2] | Nb.(RC)[1] | $HR_{5,2}$ CoI95][2] | Nb.(RC)[1] | $HR_{5,3}$ CoI95][2] | CV[3] |
| 1 | Emergence (SE) | 10 (0) | 832.1 [694.6; 1165] | 10 (0) | 1581 [1245; 2211] | 5 (4) | 2260 [NA; NA] | NA |
| 1 | Survival (SE) | 10 (0) | 177.7 [98.08; 460.6] | 10 (0) | 172.2 [93; 465.5] | 8 (2) | 146.4 [78.27; 412.6] | 0.6200 |
| 1 | Shoot dry weight (SE) | 10 (0) | 126.4 [84.87; 257.1] | 10 (0) | 125.6 [84.2; 259.5] | 9 (1) | 120.5 [80.6; 253.8] | 0.3216 |
| 1 | Survival (VV) | 10 (0) | 787 [703.1; 1069] | 10 (0) | 1184 [922.6; 2458] | 3 (3) | 8697 [NA; NA] | NA |
| 1 | Shoot dry weight (VV) | 10 (0) | 146.6 [79.96; 375.9] | 10 (0) | 142.5 [76.49; 415.4] | 9 (3) | 130.9 [70.93; 349.3] | 0.5326 |
| 2 | Emergence (SE) | 10 (0) | 41.82 [20.37; 151.6] | 10 (0) | 68.16 [32.16; 436.9] | 3 (3) | 721.9 [NA; NA] | NA |
| 2 | Survival (SE) | 10 (0) | 18.25 [6.061; 165] | 10 (0) | 18.89 [5.698; 115] | 5 (3) | 9.66 [2.666; 53700] | 2.8780 |
| 2 | Shoot dry weight (SE) | 10 (0) | 16.77 [5.313; 76.45] | 10 (0) | 17.62 [5.376; 90.29] | 8 (5) | 12.17 [2.489; 6972] | 5.2380 |
| 2 | Survival (VV) | 10 (0) | 169.5 [169; 271.5] | 10 (0) | 486.1 [NA; NA] | 1 (1) | NA [NA; NA] | NA |
| 2 | Shoot dry weight (VV) | 10 (0) | 193.3 [159.3; 259.4] | 10 (0) | 198.7 [173.5; 501.6] | 2 (1) | NA [NA; NA] | NA |
| 3 | Emergence (SE) | 10 (0) | 1495 [1343; 2199] | 10 (0) | 3049 [NA; NA] | 2 (2) | 6172 [NA; NA] | NA |
| 3 | Survival (SE) | 10 (0) | 985.2 [712; 1773] | 10 (0) | 937.6 [718.9; 2487] | 4 (1) | 902.4 [707.7; 60040] | 3.6700 |
| 3 | Shoot dry weight (SE) | 10 (0) | 278.5 [135; 936.6] | 10 (0) | 257 [129.6; 858.5] | 7 (1) | 220.8 [105.5; 899.9] | 0.6879 |
| 3 | Survival (VV) | 10 (0) | 998.9 [745.5; 1848] | 10 (0) | 973.2 [756.5; 2929] | 4 (1) | 970.3 [752.5; 3160] | 0.4557 |
| 3 | Shoot dry weight (VV) | 10 (0) | 156.5 [88.91; 410.6] | 10 (0) | 158.4 [90.34; 419] | 10 (3) | 135 [71.15; 360.8] | 0.4794 |
| 4 | Emergence (SE) | 10 (0) | 40.25 [5.791; 402.3] | 10 (0) | 50.42 [7.072; 644.8] | 6 (6) | 304600 [NA; NA] | NA |
| 4 | Survival (SE) | 10 (0) | 23.15 [4.837; 180.5] | 10 (0) | 25.9 [5.794; 199.2] | 10 (6) | 48.58 [26.2; 238.3] | 5.4630 |
| 4 | Shoot dry weight (SE) | 10 (0) | 12.3 [4.498; 41.74] | 10 (0) | 12.95 [3.311; 44.09] | 10 (1) | 25.25 [16.96; 53.34] | 0.3277 |
| 4 | Survival (VV) | 10 (0) | 71.47 [38.53; 188.6] | 10 (0) | 70 [35.79; 213.8] | 9 (4) | 57.05 [26.1; 243.4] | 3.2630 |
| 4 | Shoot dry weight (VV) | 10 (0) | 3.91 [2.018; 12.52] | 10 (0) | 3.944 [2.062; 12.3] | 10 (1) | 3.799 [1.681; 11.74] | 0.5550 |
| 5 | Emergence (SE) | 10 (0) | 0.3523 [0.1793; 1.282] | 10 (0) | 0.3731 [0.1895; 1.332] | 9 (3) | 0.3297 [0.1527; 1.594] | 70.6900 |
| 5 | Survival (SE) | 10 (0) | 0.3586 [0.1488; 1.138] | 10 (0) | 0.3836 [0.1551; 1.343] | 10 (3) | 0.3415 [0.1174; 1.317] | 0.6793 |
| 5 | Shoot dry weight (SE) | 10 (0) | 0.3437 [0.1509; 1.207] | 10 (0) | 0.3655 [0.1665; 1.297] | 10 (1) | 0.3405 [0.1427; 1.241] | 0.6585 |
| 5 | Survival (VV) | 10 (0) | 17.91 [13.86; 24.83] | 10 (0) | NA [NA; NA] | 3 (2) | 21.06 [21.06; 28.58] | 0.1296 |
| 5 | Shoot dry weight (VV) | 10 (0) | 12.24 [7.933; 21.78] | 10 (0) | 15.22 [10.38; 49.8] | 4 (2) | NA [NA; NA] | NA |
| 6 | Emergence (SE) | 10 (0) | 115.2 [54.57; 327.4] | 10 (0) | 113.2 [53.84; 574.3] | 3 (2) | 86.47 [47.41; 3535] | 1.2790 |
| 6 | Survival (SE) | 10 (0) | 29.9 [13; 121.2] | 10 (0) | 29.99 [12.3; 135.9] | 6 (2) | 21.32 [8.864; 133.9] | 2.1450 |
| 6 | Shoot dry weight (SE) | 10 (0) | 7.416 [4.924; 16.11] | 10 (0) | 7.47 [5.008; 16.77] | 10 (2) | 6.499 [3.96; 14.16] | 0.3548 |
| 6 | Survival (VV) | 10 (0) | 22.5 [14.43; 43.68] | 10 (0) | 22.53 [13.19; 50.85] | 6 (2) | 16.88 [9.297; 55.46] | 0.5118 |
| 6 | Shoot dry weight (VV) | 10 (0) | 3.525 [2.454; 7.914] | 10 (0) | 3.553 [2.449; 8.065] | 10 (1) | 3.487 [2.414; 8.764] | 0.4082 |
| 7 | Emergence (SE) | 10 (0) | 73.88 [61.31; 103.6] | 10 (0) | 151.7 [131.5; 235.4] | 3 (3) | 302.4 [NA; NA] | NA |
| 7 | Survival (SE) | 10 (0) | 7.758 [1.734; 67.84] | 10 (0) | 7.658 [1.659; 76.36] | 7 (2) | 5.275 [0.9925; 80.1] | 1.7270 |
| 7 | Shoot dry weight (SE) | 10 (0) | 1.062 [0.2098; 6.958] | 10 (0) | 1.553 [0.3054; 6.965] | 10 (1) | 1.401 [0.2398; 6.885] | 0.8095 |
| 7 | Survival (VV) | 10 (0) | 6.341 [3.258; 22.57] | 10 (0) | 6.516 [3.28; 25.68] | 7 (3) | 5.612 [2.314; 35.4] | 4.6280 |
| 7 | Shoot dry weight (VV) | 10 (0) | 1.448 [0.7201; 4.423] | 10 (0) | 1.402 [0.7252; 4.51] | 10 (2) | 1.152 [0.496; 3.481] | 0.5719 |

NA stands for Not Available; it may appear either when there is a problem of convergence, or when the proximity or the equality of $ER_{50}$ values leads to always bootstrapping the same set of $ER_{50}$ values thus providing equal lower and upper bounds of the $CoI95$. Lines coloured in gray stand for $HR_{5,3}$ poorly estimated or not-estimated based on less than six distinct intervals for $ER_{50}$ inputs, or for only right-censored $ER_{50}$ as inputs for SSD analyses.

[1] Number of distinct $ER_{50}$ (number of distinct right-censored $ER_{50}$);

[2] Estimated $HR_5$ [95% confidence interval];

[3] Coefficient of variation for $HR_5$.

where the $HR_{5,3}$ was greater than the $HR_{5,1}$ and $HR_{5,2}$ estimates, the right-censored $ER_{50}$ values were not only obtained for less sensitive species but also for highly sensitive species (i.e., species having lower median $ER_{50}$ values). This happened often in case studies for which there were lots of right-censored $ER_{50}$ values, as the application rates were not adapted for most of the

chosen species for the measured endpoints. In a risk assessment context, this situation should typically lead to new experiments with higher tested rates.

## Discussion

### Dose-response modelling

**Model choice.**   There are a vast variety of models in common use to describe a dose-response relationship for ecotoxicity test data, such as probit, log-logistic, Weibull, etc. [16–18]. The log-logistic models have been widely used in weed science and they have been recommended as a standard herbicide dose-response [19]. The log-logistic models are by far the most commonly used model for describing toxicity test data [20]. The log-logistic models can be used to properly analyse not only continuous data but also quantal data. Hence, for our NTTP data, we always used log-logistic models with the same deterministic part to analyse emergence, survival and shoot dry weight data, thus facilitating comparisons. Hence a three-parameters log-logistic model was chosen to analyse not only emergence and survival data, but also for shoot dry weight data. Although a four-parameters log-logistic model could have been tested and may be chosen for shoot dry weight data, we preferred to use the three-parameters one, since it is reasonable to fix the lower asymptote (parameter c) at 0 considering that, at really high application rates all plants can die, even though the actual data are not fully supporting this assumption for few rare cases [20]. Morever, for any of our data sets, the addition of one extra parameter did not significantly improve the model fitting (results not shown).

**Choice of priors.**   A quasi-non-informative uniform prior distribution was chosen for the logarithm of parameter $b$ within the interval $[−2, 2]$, in order to cover a wide variety of dose-response shapes. In certain cases, we had an extremely flat dose-response curve (no effect at any of the tested rates was observed on the endpoint) so that the posterior of parameter $b$ was as wide as the prior, even if we enlarged the support prior interval. In addition, in these cases, the imprecise estimation of parameter $b$ did not influence our conclusions on the $ER_{50}$ estimates, since the $ER_{50}$ values were considered as right-censored with their lower bound being the highest tested rate.

For parameter $e$, we used the same prior as the one used in the R-package `morse` [12, 21]: a normal distribution was chosen for the logarithm of parameter $e$, with specific mean and standard deviation (presented in section method) computed from the experimental design. This choice implies that parameter $e$ has a probability slightly greater than 95% to lie within the range of tested rates.

A uniform prior distribution was chosen for parameter $d$ within the interval $[0, d_{max}]$. For the emergence and survival data sets, $d_{max}$ equals 1 representing a 100% probability of emergence or a 100% probability of survival. For the shoot dry weight data sets, $d_{max}$ should ideally be chosen according to expert knowledge and equal to the highest expected shoot dry weight regarding the species and environmental conditions (in the experiment) under consideration. Nevertheless, for pratical convience, $d_{max}$ was chosen as twice the highest observed shoot dry weight for the species under consideration, and then the observation with the highest observed value was discarded from the dose-response analyses.

Concerning prior distributions for the variance parameter of the Gaussian distribution (shoot dry weight data only), there are commonly used prior specifications (*e.g.*, an inverse gamma, an inverse chi-square distribution on variance parameter, a uniform distribution on standard deviation parameter). We finally assigned a uniform prior distribution to standard deviation parameter $\sigma$ within the interval $\left[0, \frac{d_{max}}{2}\right]$.

## Censoring of $ER_{50}$ estimates

We chose to use criterion $C_1$ based on the ratio of probabilities and a T value equal to 0.5 to automatically censor $ER_{50}$ estimates. Indeed, criteria $C_1$ and $C_2$ led to very close censoring decisions for $ER_{50}$ estimates, but criterion $C_1$ was preferred since it is based on the whole probability distribution of the $ER_{50}$. However, if criterion $C_2$ had been chosen for censoring $ER_{50}$ estimates, there would be very few changes for SSD analyses and $HR_5$ estimates. Regarding decision threshold T, there is no rule set in stone for its choice. We have a preference for T = 0.5, as a neutral value. This medium value may avoid considering very imprecise $ER_{50}$ estimates as interval-censored by their $CI95$ and may also avoid considering enough precise $ER_{50}$ estimates as right-censored. Based on the 350 data sets we analysed, the same results and conclusions are almost always reached for a choice of a T value between 0.4 and 0.6. We propose to always use the intermediate value of 0.5 for the sake of simplicity and comparability. We recommend to avoid using T values below 0.4 or above 0.6, for the reasons mentioned above.

**Uncertainty on the $ER_{50}$ estimates.** There are some limitations in the way we considered the uncertainty on the $ER_{50}$ estimate. Indeed, the uncertainty was simply summarised by either an interval-censored $ER_{50}$ with its $CI95$ or by a right-censored $ER_{50}$ accounting for the range of tested rates. This method does not use all the available information on $ER_{50}$ (*i.e.*, the full posterior distribution). It is conceivable that the uncertainty on the $ER_{50}$ estimates could be considered in other better ways, allowing to account for the full posterior distribution of $ER_{50}$ within subsequent SSD analyses. Further research would be needed to explore this possibility.

## SSD analyses

Compared to the traditional deterministic approach that relies on the most sensitive individual toxicity data, the probabilistic SSD method has numerous advantages [2, 22]. As always in statistics, SSD can be built with either parametric or non-parametric methods. Some parametric distributions have already been proposed for SSD, such as log-logistic [23], log-normal [3, 24–28], Burr Type III [29], Weibull distributions, etc. The common use of the parametric approach for SSD is due to its mathematical simplicity and because it requires less data points compared to non-parametric approaches. Log-normal and log-logistic distributions are the most commonly used for SSD [3, 26, 27]. The present paper used a log-normal distribution to fit $ER_{50}$ values without testing the normality of the logarithm of tested rates, since it is not our purpose to find the best fit to toxicity values, but rather to study the influence of accounting for the uncertainty of $ER_{50}$ inputs on $HR_5$ estimates.

Concerning minimum data requirements for fitting an SSD, a minimum of five to ten species is deemed acceptable for regulatory purposes in aquatic ecotoxicity [2] and, in the context of environmental risk assessment, a minimum of six species is required, ten being usually recommended [6]. Indeed, small size of input samples may lead to high uncertainty in fitted SSD [26]. Ten data points were also suggested by Wheeler et al. [30] to generate reliable estimate upon which regulatory decisions may be based. In the present paper, ten NTTP species were therefore tested for the seven case studies, allowing to collect ten $ER_{50}$ values (generally distinct) for SSD analyses. However, for certain endpoints of certain case studies, when taking into account right-censored $ER_{50}$ values in SSD analyses, the number of distinct $ER_{50}$ values for SSD diminished, since some of the right-censored values were in fact equal to the same interval [*max_rate*, $+ \infty$], with *max_rate* equal for all the species. Consequently, we had some cases where SSD were poorly fitted with less than six distinct intervals for $ER_{50}$ values, some of which being right-censored. When this happens, it is better not to consider SSD results and their corresponding estimated $HR_5$ values, and if possible to add new experiments with higher application rates. Based on our seven cases studies, at least six distinct intervals (whatever the

$ER_{50}$ point values) appears as the minimum requirement to reasonably fit an SSD and obtain a relevant 95% confidence interval on the final $HR_5$ estimate.

**Influence of right-censored data on $HR_5$ estimates.** For endpoints for which the SSD was fitted on at least six distinct intervals for $ER_{50}$ inputs, most of the time, $HR_5$ estimates taking the right-censoring into account were lower than $HR_5$ estimates based on medians of $ER_{50}$ or interval-censored $ER_{50}$ by their $CI95$. The results were consistent with a simulation study conducted by Green et al. [16], which demonstrated that the mathematically sound way of using censored data tends to underestimate $HR_5$ compared to the $HR_5$ resulting from a data set without censored values, with greater underestimation associated with greater percentage of censoring. In addition, we found that, when right-censored $ER_{50}$ values were obtained for the less sensitive species, which is the most common case in practice, the $HR_5$ values obtained by including the right-censoring were most of the time smaller than those obtained by handling censored data as non-censored data.

On a general point of view, based on our seven case studies for the five endpoints we analysed, the influence of including right-censored data on the $HR_5$ estimate depends on the right-censored $ER_{50}$ values being obtained rather on more sensitive species or on less sensitive species. In addition, we can say that, if right-censored data spread in a random way among the chosen species, the $HR_5$ obtained by considering right-censored $ER_{50}$ values can be both greater or smaller than the $HR_5$ obtained by handling censored data as non-censored ones.

**Sensitivity of endpoints.** In the result section, we have not mentioned the sensitivity of endpoints. However from Table 3, we found that the shoot dry weight endpoint from SE or VV tests appeared almost always as the most sensitive endpoint with the lowest estimated $HR_5$ value, except for case study 5 for which $HR_5$ results for three endpoints (emergence, survival and shoot dry weight of the SE test) were very close. The shoot dry weight from the VV test appeared more often as the most sensitive one (four times out of the seven case studies) than the one from the SE test. In addition, for the shoot dry weight endpoint, we had rarely right-censored $ER_{50}$ values according to C1T0.5. Therefore, it would be recommended to always collect and analyse shoot dry weight data in order to assess risk of herbicide on NTTP by using SSD analyses based on censored $ER_{50}$ inputs.

**Experimental design.** In certain studies, the tested rates were not specifically adapted to the sensitivity of some species. Hence, some species were not affected or slightly affected at the highest intended application rate, leading to right-censored $ER_{50}$ values. These right-censored values may affect the estimation of $HR_5$, for example making the estimate less precise. If the precision of the $HR_5$ is not considered as sufficient (*i.e.*, with a high value of the CV), then conducting new experiments with higher application rates may help to refine the final estimation of the $HR_5$.

## Conclusion

All our results confirm the usefulness of our integrated approach going from raw toxicity test data until the $HR_5$ (or $HC_5$) estimation, considering uncertainty propagation all along the data analysis process. Accounting for $ER_{50}$ (or $EC_{50}$) estimates as intervals clearly avoid to discard any inputs for SSD analyses, or to arbitrarily convert them to point values. This also avoid to increase uncertainty in the apical estimate of the $HR_5$ (or $HC_5$) by keeping as much $ER_{50}$ (or $EC_{50}$) inputs as possible whatever their associated type of interval (bounded or not). Additionally, the method we proposed in our paper is applicable to any taxon in ecotoxicology. The results we presented are based on a total 350 data sets consisting of seven case studies, each with five endpoints (survival, emergence, shoot dry weight) for 10 non-target terrestrial plants from standard greenhouse experiments that used different experimental designs (vegetative

vigour vs. seedling emergence) and applied herbicides at different growth stages. The Bayesian framework allowed estimating $ER_{50}$ values and 95% credibility intervals for all data sets, even when the dose-response curve did not reach a strong effect at the highest tested rate. Combined with a statistically sound approach for inclusion of censored $ER_{50}$ estimates in SSD computing, we maximised the use of existing species data when building SSD, thus avoiding discarding right and/or left-censored data that may be obtained from lab studies for less or more sensitive species. Our overarching study confirmed that at least six distinct intervals (whatever the $ER_{50}$ point values) are required as input to the SSD analysis to ensure obtaining a reliable estimate of the $HR_5$.

Our paper finally proposes a statistically sound method for propagating the uncertainty of the $ER_{50}$ (or $EC_{50}$) estimates considered as interval-censored values towards the $HR_5$ (or $HC_5$) estimates. This method delivers both point estimates and bootstrap 95% confidence intervals of $HR_5$ (or $HC_5$). It illustrates that both propagating the uncertainty from $ER_{50}$ (or $EC_{50}$) estimates and including interval-censored data as inputs for SSD analyses may change both the point estimate and the 95% confidence interval on the $HR_5$ (or $HC_5$). The extend of the change depends on the characteristics of the $ER_{50}$ (or $EC_{50}$) input values (*e.g.*, whether censored data were obtained for less or more sensitive species or were randomly spread among the tested species) and on the chosen criteria for handling the uncertainty of $ER_{50}$ (or $EC_{50}$) values. Consequently, when comparing and interpreting the final results, we recommend not to focus solely on the point estimate of the $HR_5$ (or $HC_5$), but also to look at the precision of this estimate as depicted by its 95% confidence interval. A small confidence interval stands for a precise estimate of the $HR_5$ (or $HC_5$), and thus a low uncertainty. This information integrates both the differences in sensitivity and the uncertainty of the $ER_{50}$ (or $EC_{50}$) inputs across a range of species all the way down to the $HR_5$ (or $HC_5$) estimation: therefore, it is particularly valuable for an informed use of the $HR_5$ (or $HC_5$) value in the context of environmental risk assessment.

## Supporting information

**S1 Table. Corresponding species and common names to the EPPO code.**
(PDF)

**S2 Table. Formulation of active substances in seven products.**
(PDF)

**S1 Appendix. Reproduction of results via MOSAIC.**
(PDF)

**S1 Archive. Zip file containing all supplementary results.** It is a zip file containing seven folders (one folder per case study). Each folder contains five files `report_xxx.pdf` with detailed results of the dose-response analyses, one file corresponding to does-response analysis per endpoint. It also contains one file `ER50_censoring.pdf` for censored $ER_{50}$ and one file `SSD_analyses.pdf` for results of SSD analyses.
(ZIP)

## Acknowledgments

The authors are particularly indebted to Aude RATIER and Gauthier MULTARI who made significant improvements in the web tool MOSAIC_growth associated to dose-response analyses we performed on growth-type data (namely shoot dry weight data from SE and VV toxicity tests): https://mosaic.univ-lyon1.fr/growth.

## Author Contributions

**Conceptualization:** Sandrine Charles, Virginie Ducrot.

**Data curation:** Dan Wu, Virginie Ducrot.

**Formal analysis:** Dan Wu.

**Funding acquisition:** Sandrine Charles, Virginie Ducrot.

**Methodology:** Sandrine Charles, Dan Wu.

**Project administration:** Sandrine Charles, Virginie Ducrot.

**Software:** Dan Wu.

**Supervision:** Sandrine Charles, Virginie Ducrot.

**Writing – original draft:** Sandrine Charles, Dan Wu, Virginie Ducrot.

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
