## [Decision Letter · Decision Letter 0]

19 Nov 2020

PONE-D-20-21130

How to account for the uncertainty from standard toxicity tests in species sensitivity distributions: an example in non-target plants

PLOS ONE

Dear Dr. CHARLES,

Thank you for submitting your manuscript to PLOS ONE. After careful consideration, we feel that it has merit but does not fully meet PLOS ONE’s publication criteria as it currently stands. Therefore, we invite you to submit a revised version of the manuscript that addresses the points raised during the review process.

We look forward to receiving your revised manuscript.

Kind regards,

Mohammad Ansari

Academic Editor

PLOS ONE

Journal Requirements:

2.Thank you for stating the following in the Financial Disclosure section:

[The European Crop Protection Association (ECPA, https://www.ecpa.eu) financially supported this work. The funder had no role in study design, data collection and analysis, decision to publish, or preparation of the manuscript.].   

We note that one or more of the authors are employed by a commercial company: Bayer AG, Crop Science,

Reviewers' comments:

Reviewer's Responses to Questions

**Comments to the Author**

1. Is the manuscript technically sound, and do the data support the conclusions?

Reviewer #1: No

Reviewer #2: Yes

2. Has the statistical analysis been performed appropriately and rigorously? 

Reviewer #1: Yes

Reviewer #2: Yes

3. Have the authors made all data underlying the findings in their manuscript fully available?

Reviewer #1: Yes

Reviewer #2: Yes

4. Is the manuscript presented in an intelligible fashion and written in standard English?

Reviewer #1: Yes

Reviewer #2: Yes

5. Review Comments to the Author

Reviewer #1: Manuscript covers the content which justify the title. Toxicity modelling described in a compiled form which makes the manuscript suitable to be published in this journal. I appreciate the work of the authors.

Reviewer #2: The authors have presented a novel methodology, but few comments are still remaining as below:

1- What is the novelity of this study?

2- What is the difference/ advancement in this methodology with the other ones present?

3- What is the usefulness of this methodology?

6. PLOS authors have the option to publish the peer review history of their article (what does this mean?). If published, this will include your full peer review and any attached files.

Reviewer #1: **Yes: **Amit Verma

Reviewer #2: No

---

## [Author Response · Author response to Decision Letter 0]

15 Dec 2020

Our respond to reviewers is included within our cover letter.

---

## [Decision Letter · Decision Letter 1]

22 Dec 2020

How to account for the uncertainty from standard toxicity tests in species sensitivity distributions: an example in non-target plants

PONE-D-20-21130R1

Dear Dr. CHARLES,

We’re pleased to inform you that your manuscript has been judged scientifically suitable for publication and will be formally accepted for publication once it meets all outstanding technical requirements.

Kind regards,

Mohammad Ansari

Academic Editor

PLOS ONE

Additional Editor Comments (optional):

Reviewers' comments:

Reviewer's Responses to Questions

**Comments to the Author**

1. If the authors have adequately addressed your comments raised in a previous round of review and you feel that this manuscript is now acceptable for publication, you may indicate that here to bypass the “Comments to the Author” section, enter your conflict of interest statement in the “Confidential to Editor” section, and submit your "Accept" recommendation.

Reviewer #1: All comments have been addressed

Reviewer #2: All comments have been addressed

2. Is the manuscript technically sound, and do the data support the conclusions?

Reviewer #1: Yes

Reviewer #2: Yes

3. Has the statistical analysis been performed appropriately and rigorously? 

Reviewer #1: Yes

Reviewer #2: Yes

4. Have the authors made all data underlying the findings in their manuscript fully available?

Reviewer #1: Yes

Reviewer #2: Yes

5. Is the manuscript presented in an intelligible fashion and written in standard English?

Reviewer #1: Yes

Reviewer #2: Yes

6. Review Comments to the Author

Reviewer #1: Manuscript drafted well to cover all the aspects related to the title. ..............................

Reviewer #2: The authors have now detailed their study in the revised manuscript. The manuscript seems good in its present form. I accept the authors comments

7. PLOS authors have the option to publish the peer review history of their article (what does this mean?). If published, this will include your full peer review and any attached files.

Reviewer #1: **Yes: **Amit Verma

Reviewer #2: No

---

## [Editor Report · Acceptance letter]

28 Dec 2020

PONE-D-20-21130R1 

How to account for the uncertainty from standard toxicity tests in species sensitivity distributions: an example in non-target plants 

Dear Dr. Charles:

I'm pleased to inform you that your manuscript has been deemed suitable for publication in PLOS ONE. Congratulations! Your manuscript is now with our production department. 

Kind regards, 

on behalf of

Dr. Mohammad Ansari 

Academic Editor

PLOS ONE